# Four-party evolutionary game analysis of enterprise environmental behavior

**Xujun Zhai[1], Lian Zheng[2], Hong Lin[3]***

1 New Huadu Business School, Minjiang University, Fuzhou, China, 2 College of Economics and Management, Heilongjiang Bayi Agricultural University, Daqing, China, 3 College of Marxism, Minjiang University, Fuzhou, China

* linhong_1977@163.com

## Abstract

With the implementation of the "Rural Revitalization Strategy" in China, it is common for enterprises to go to the countryside to develop business. However, enterprises often neglect the local environmental protection in rural areas while developing the economy to pursue profits. As the end of the national administrative system and the villagers' autonomous organization, the village committee needs to participate in monitoring enterprises' environmental behavior. With this in mind, this paper builds a game model of enterprises, grass-roots governments, farmers, and village committees and analyzes the impact of village committees, grass-roots governments, and farmers on enterprise environmental behavior. The conclusions are as follows: (i) it is difficult for the village committee to promote the positive environmental behavior of enterprises, which needs the supervision of the grass-roots government; (ii) Improving the coordination ability of village committees is conducive to reducing the burden of government supervision; (iii) Farmers' awareness of environmental protection can affect the environmental behavior of enterprises through the rights protection mechanism and reputation mechanism.

## 1 Introduction

With the implementation of the "Rural Revitalization Strategy" in China, many production resources and policy dividends are tilted to the countryside [1], which has become a "blue ocean" for enterprises to enter, and it is common for enterprises to go to the countryside to develop business. However, when enterprises go to the countryside to promote the upgrading and development of rural industries, all kinds of pollution also pose a significant threat to the rural environment. In recent years, "beautiful country" has become one of the objectives of the Rural Revitalization Strategy, and the state has highly valued rural environmental issues. China's "No. 1 central document" for 2016 indicated that the rural environmental improvement expenditure should be gradually included in the local financial budget. "No. 1 central document" for 2018 proposed to implement the three-year action plan for improving rural human settlements, making specific plans for improving the rural environment. "No. 1 central document" for 2022 successively proposed a five-year action plan, taking into account the needs of

**Data Availability Statement:** All relevant data are within the manuscript and its Supporting Information files.

**Funding:** Xujun Zhai, National Social Science Foundation of China (19BJL054). The funders had

no role in study design, data collection and analysis, decision to publish, or preparation of the manuscript.

**Competing interests:** The authors have declared that no competing interests exist.

farmers and regional differences, consolidating the achievements of environmental remediation, and further promoting the treatment of water bodies, and domestic waste. However, in reality, rural ecological problems are still prominent, and the parallel between enterprises and pollution going to the countryside has further aggravated the problematic situation of the rural environment, such as soil acidification, water pollution, and biodiversity reduction caused by the abuse of chemical drugs [2]. Therefore, standardizing the environmental behavior of enterprises after going to the countryside is an inevitable requirement to consolidate the achievements of rural environmental remediation and achieve the strategic goal of a "beautiful country."

With the proposal of the diversified governance model for the rural environment, the village committee has become one of the main governance subjects mentioned several times because of its administrative role and autonomy. As the end of the national organizational system and the villagers' autonomous organization, the village committee not only undertakes the functions and powers of protecting and improving the rural ecological environment stipulated by the state but is also responsible for the environmental interest demands expressed by the state and the villagers. This dual identity jointly determines that the village committee will become the inevitable main body when enterprises go to the countryside to cause environmental pollution, infringe on villagers' environmental interests, and participate in the impact process of enterprise environmental behavior. Therefore, this paper adds the village committee as the main body, establishes a dynamic game model among the village committee, grass-roots government, enterprises, and farmers, and studies the impact of the village committee, grass-roots government, and farmers on enterprise environmental behavior.

## 2 Literature review

### 2.1 Research on the influence of government on enterprise environmental behavior

Previous researches show there are two viewpoints about the influence of government on enterprise environmental behavior:

1. Partial research suggested that the absence of government supervision led to widespread environmental damage behaviors of enterprises. For example, Hu (2021) carried out an empirical study, and the results showed that compared with cities without government supervision, SO2 emissions of enterprises increased by 11.3% [3]; Yu (2019) pointed out that the evaluation system of local officials makes local governments ignore environmental issues, and even collude with local enterprises to achieve economic growth, resulting in environmental pollution [4].

2. Partial research believed that too strict government supervision had frustrated the enthusiasm of enterprises for environmental protection. For example, Xie (2017) thought that for most provinces in China, the current strictness degree of supervision is reasonable, and strengthening leadership within the best strictness can improve environmental quality [5]; Liao (2018) emphasized that the government's environmental policy is an essential means to promote enterprise ecological innovation, and further analyzed the impact of three environmental policy tools, command and control, marketization, and informatization, on enterprise environmental innovation [6].

Farida (2020) tested the impact of stakeholders on the environmental behavior of enterprises in the context of emerging economies. The results showed that regulators are the most critical stakeholders, and the environmental policies of enterprises are required by government

supervision [7]. Peng (2021) pointed out that the government's environmental supervision measures should guide enterprises to intimate environmental behavior. At the same time, the government should not only pay attention to monitoring enterprise pollution emissions but also to enterprise green innovation [8].

## 2.2 Research on the influence of the public on enterprise environmental behavior

Previous research mainly used two types of methods to test the influence of the public on enterprise environmental behavior.

1. Some studies use econometric methods to analyze the impact mechanism of public participation on enterprise environmental behavior. For example, Han (2016) took enterprise ethics as the intermediary variable. He used the single-factor variance analysis method to verify the impact of enterprise environmental pollution on its reputation and the role of public awareness of enterprise ethics [9]. Wang (2020) constructed a multiple regression model to empirically study public attention's positive impact on enterprise environmental information disclosure [10].

2. Some studies used game theory to analyze the interaction between public participation, government supervision, enterprise emissions, and other subjects' strategic choices. For example, Chen (2020) built a tripartite game model between the government, manufacturers, and the public under carbon taxes and subsidies. The analysis found that public participation can help the government supervise the manufacturers' production to compensate for the lack of government supervision [11]. You et al. (2017) proposed that the government can improve the design of environmental regulation by increasing the public's participation based on the evolution game model [12].

## 2.3 Research on the village committee's response to rural environmental problems

As for how the village committee deals with rural environmental pollution, the existing literature often gives countermeasures and suggestions on how to deal with rural environmental pollution based on the role of the village committee. The role of the village committee can be divided into three categories: (i) The agent of the government, whose main responsibility is to manage village affairs; (ii) The agent of the villagers to express the interests of the villagers to the government; (iii) The intermediary between the government and the villagers plays a two-way communication role. For example, Tang et al. (2020) explored the significant role that village committees played in farmers' withdrawal from rural homesteads [13]; Hu and Kee (2022) revealed the improvement measures for village officials' corruption prevention in Chinese village residents' committee [14].

The existing literature primarily focuses on the impact between the government, the public, and enterprises or the game analysis between the three parties, emphasizing the significant impact of the government and the public on the environmental behavior of enterprises. However, few works of literature divide the research area into urban and rural areas, which leads to the neglect of the village committee, a particular governance body in rural areas. In the rural area, the village committee, as an autonomous organization of villagers, has unique geographical advantages in controlling the environmental pollution of enterprises and plays an irreplaceable primary role. Therefore, in the specific situation of rural environmental pollution caused by enterprises going to the countryside, this paper brings the village committee into the scope

of the main body, constructs a game model of the four central bodies of the village committee, farmers, grass-roots government and enterprises, and discusses the impact of the village committee, farmers and grass-roots government on the environmental behavior of enterprises.

## 3 Methodology

The evolutionary game model is the leading research method in this paper. The interests of each subject are constantly changing with the change in Rural Revitalization Strategy. At the same time, all subjects will constantly adjust their strategies according to the strategies of other subjects [15]. In the evolutionary game theory, the derivation process of the evolutionary game model is described in detail. On this basis, this paper will substitute the subjects involved in the "Rural Revitalization Strategy" into the model to obtain the game relationship between them.

Farmers are the beneficiaries of rural revitalization, and the purpose of rural development is to improve farmers' income. Increasing rural income needs to carry out industrial upgrading, and enterprises can use their capital, technology, and other advantages to drive the development of rural industries and help implement the "Rural Revitalization Strategy" smoothly [16]. However, due to bounded rationality, enterprises and farmers may have speculative behavior, and farmers are more likely to infringe on their interests because of their weak personal strength. Therefore, the interaction between enterprises and farmers cannot be separated from the supervision of the government, especially the grass-roots government [17]. In addition, as an autonomous grass-roots organization of a mass character, the village committee is the main body with the most direct contact with rural production and life. It plays a dual role of speaking on behalf of rural public opinion and implementing township government policies. It can be seen that farmers, grass-roots governments, enterprises, and village committees are the four essential subjects in the process of rural environmental protection. Based on this, this paper adopts the method of the evolutionary game. It supposes that farmers, grass-roots governments, enterprises, and village committees are rational subjects to maximize their interests and constantly adjust and improve their strategic choices according to their benefits in the game process.

## 4 Model development

### 4.1 Model framework

In the implementation process of rural revitalization, to achieve industrial upgrading, enterprises have been introduced into the countryside to use their capital and technological advantages to drive rural development. The grass-roots government will supervise the enterprise in the production process, and the grass-roots government will supervise the production safety of the enterprise according to rules and regulations [18]. However, even so, enterprises still have the possibility of pursuing high profits and illegal operations [19]. When enterprises produce nonstandard business behaviors such as wanton discharge of wastewater and waste gas, it will cause the adverse consequences of reducing farmers' economic income, damaging the rural ecological environment, declining farmers' life quality, and hindering sustainable rural development [20].

At this time, farmers with environmental awareness will actively report to the village committee when they are aware of the damage to the rural environment and expect the village committee as a representative to negotiate with the enterprises involved and pay attention to the results of the negotiation. The village committee is responsible for supervising the enterprises' production behavior, protecting the beauty of the rural environment, and safeguarding the interests of farmers. When the village committee fails to do its duty, it takes an indifferent attitude to the illegal business behavior of the enterprise. Nevertheless, when the village

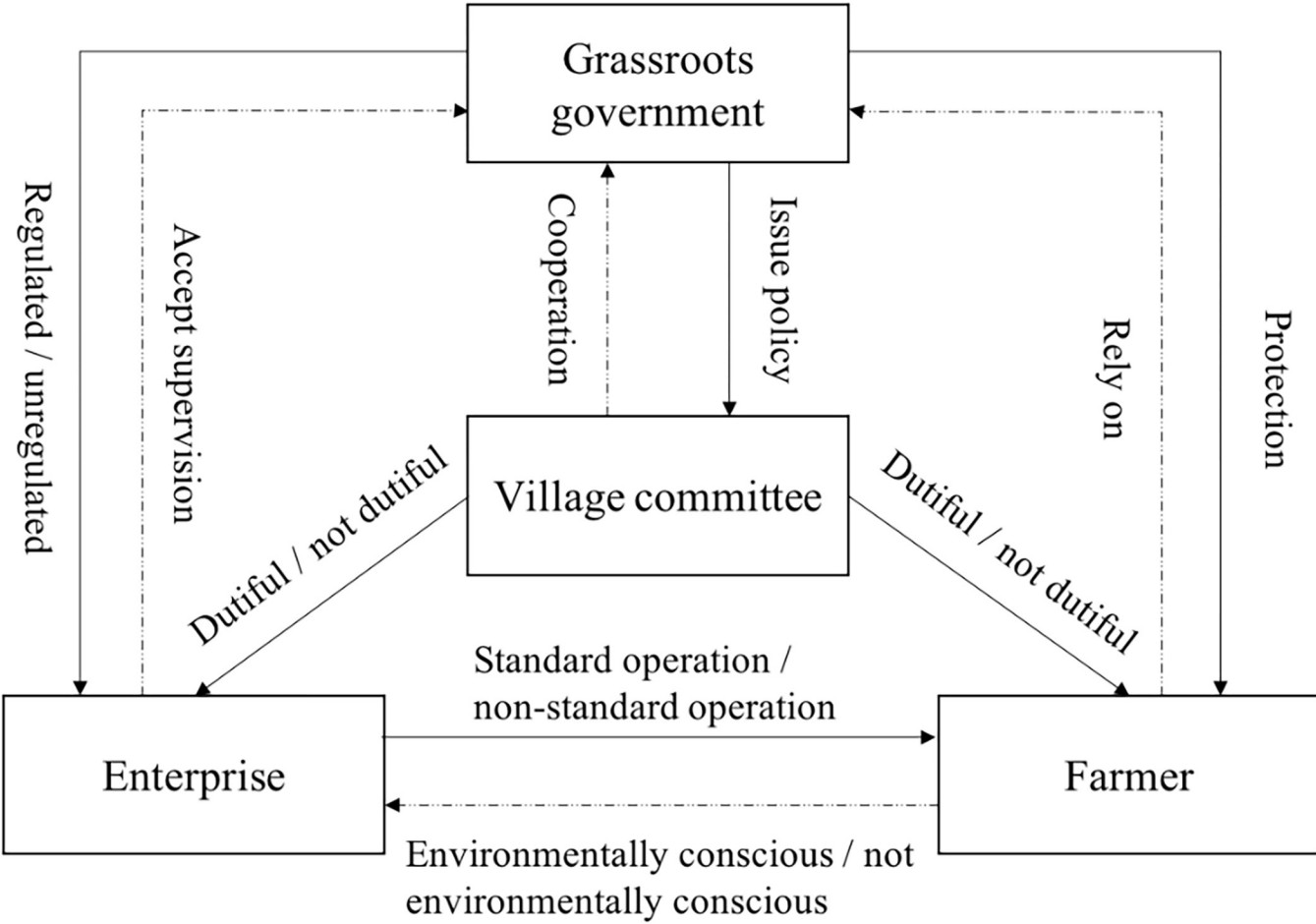

**Fig 1. Logical relationship between the four players.**

committee does its duty, whether the farmers report and reflect on the situation, it can timely find the nonstandard business behavior of the enterprise, encourage the enterprise, and coordinate the environmental interest conflict between the enterprise and the farmers. Besides, when the village committee fails to fulfill its duties or fails to coordinate, farmers are not satisfied with the results given by the village committee. They may continue to appeal to the grassroots government to safeguard their rights and finally resort to legal channels to safeguard their rights and interests.

Therefore, this paper mainly discusses the following questions: (i) can the participation of village committees improve the probability of standardized operation of enterprises and adopt positive environmental behavior? (ii) How does farmers' environmental protection awareness affect enterprises' environmental behavior decisions? This paper constructs a multi-agent game model for farmers, enterprises, village committees, and grass-roots governments. The logical relationship between the four players is shown in Fig 1:

## 4.2 Assumptions and model development

**Hypothesis 1:** there are four participants in the evolutionary game: enterprises, grass-roots governments, farmers, and village committees. Each participant is bounded rational, and the information between them is not entirely symmetrical.

**Hypothesis 2:** the enterprise has two standard and nonstandard operation strategies, with the probability of $x$ and $1-x$ respectively. The probability of farmers' environmental protection awareness is $z$. At this time, farmers are susceptible to environmental quality and can perceive a beautiful environment's physical and mental pleasure. The probability that farmers have no awareness of environmental protection is $1-z$. As long as the enterprise does not cause substantial and direct harm to themselves, they will have an indifferent attitude towards the environmental quality in the village, and small changes in environmental conditions will not bring their emotional fluctuations, let alone pay attention to the results of the village committee's negotiation with enterprises on pollution incidents. The probability of a dutiful and irresponsible village committee is $q$ and $1-q$, respectively. The probability of grass-roots government supervision is $y$, and the probability of non-supervision is $1-y.x$, $y,z,q\in[0,1]$.

**Hypothesis 3:** When the enterprise chooses to operate standardized, it will invest money to buy environmental protection equipment or choose a higher-cost production method. Currently, the cost is recorded as $C_{x1}$, and the cost of nonstandard operation is recorded as $C_{x2}$. When the enterprise does not operate in a standard way, if it accepts the coordination of the village committee or is supervised by the grass-roots government, it will invest in additional environmental protection facilities, which will be recorded as $F1$. At this time, losses suffered by the enterprise are small and the losses are considered to be 0 to simplify the calculation. If the village committee fails to coordinate with the enterprise or fulfill its duties, the government does not supervise. The enterprise will operate wantonly and nonstandard, and the environmental pollution will further deteriorate, causing significant physical and mental harm to farmers, which is recorded as $N_z$. At the same time, environmental pollution incidents will harm the smooth implementation of the "Rural Revitalization Strategy", and the resulting social losses are recorded as $N_{y\circ}$

**Hypothesis 4:** As the most direct manager of rural affairs, the dutiful village committee can find out the nonstandard production behavior of enterprises faster than the grass-roots government. The village committee will timely and effectively convey the laws, regulations, and relevant policies of the grass-roots government to enterprises, and coordinate the environmental conflicts of interest between enterprises and farmers. This cost of time and energy is recorded as $C_q$. Suppose the capacity of the village committee is $\alpha,\alpha\in[0,1]$, which indicates the probability of the enterprise accepting the coordination of the village committee and adding investment in environmental protection equipment is $\alpha$. That is, the probability of successful coordination is $\alpha$.

**Hypothesis 5:** The supervision cost is recorded as $C_y$ when the grass-roots government chooses supervision. The enterprises that fail to accept the coordination of the village committee and insist on the nonstandard operation and the village committee that fails to do their duty will be subject to economic punishment, and the fines will be recorded as $F2$ and $F4$ respectively.

**Hypothesis 6:** When farmers are aware of environmental protection, they will pay close attention to the coordination results between the village committee and the enterprise and evaluate the behavior of the village committee and the enterprise according to the coordination results. Suppose the village committee fails to take positive action on the reflected situation. In that case, farmers believe that the village committee does not fulfill its duty to maintain the beauty of the village environment, and they will reduce trust in the village committee and cooperation degree in the subsequent affairs, and the negative effect is recorded as $N_q$. For enterprises that insist on nonstandard operation, on the one hand, farmers will be

**Table 1. Parameter settings and meanings.**

| Parameter | Meaning | Parameter | Meaning |
|---|---|---|---|
| $C_{x1}$ | Cost of standardized operation of the enterprise | $C_{x2}$ | Cost of nonstandard operation of the enterprise |
| $C_y$ | Cost of government supervision | $C_q$ | Cost of fulfilling the duty of the village committee |
| $N_q$ | The dissatisfaction of farmers with environmental awareness when the village committee fails to do its duty | $N_z$ | Losses to farmers caused by enterprise adhering to nonstandard operation |
| $N_z$ | Losses to society caused by enterprise adhering to nonstandard operation | $R_x$ | Reputation loss of enterprise adhering to nonstandard operation |
| F1 | Enterprise accepts the coordination of the village committee and adds additional investment in environmental protection facilities | $R_z$ | Farmers with environmental awareness feel the physical and mental pleasure brought by the beautiful environment |
| F2 | Fines imposed by the grass-roots government on the enterprise that do not operate in a standardized manner | F3 | Economic compensation for farmers' rights protection |
| F4 | Economic fines imposed by the grass-roots government on village committees for failing to perform their duties | $T_y$ | Social losses recovered after compensation by the enterprise |
| $\alpha$ | Probability of successful coordination of village committee | $\beta$ | The probability of farmers with environmental awareness continuing to protect their rights |

dissatisfied, thinking that the enterprise is mercenary and spread word of mouth, causing the reputation loss of the enterprise, which is recorded as $R_x$. On the other hand, farmers may continue to petition for rights protection. This probability is $\beta$, to obtain economic compensation F3. This behavior also recovers the social losses of the enterprise to a certain extent, which is recorded as $T_y$.

In summary, all parameter settings and their meanings are shown in Table 1, and the game income matrix of grass-roots government, village committee, enterprise, and farmers are shown in Table 2:

The formulas in Table 2 are based on the strategic interactions between enterprises, village committees, grass-roots government, and farmers. They incorporate costs, fines, probabilities, and the impacts of environmental awareness to derive expected payoffs in various scenarios. The formulas in Table 2 are derived by considering the strategic interactions between each

**Table 2. Game income matrix of grass-roots government, village committee, enterprise, and farmers.**

| Enterprise | Village committee | Grass-roots government | | | |
|---|---|---|---|---|---|
| | | Supervision $y$ | | Non-supervision $1-y$ | |
| | | Farmers have environmental awareness $z$ | Farmers have no environmental awareness $1-z$ | Farmers have environmental awareness $z$ | Farmers have no environmental awareness $1-z$ |
| Standardized operation $x$ | Dutiful $q$ | $-C_{x1}$ $-C_y$ $R_z$ $-C_q$ | $-C_{x1}$ $-C_y$ 0 $-C_q$ | $-C_{x1}$ 0 $R_z-C_q$ 0 | $-C_{x1}$ 0 0 $-C_q$ |
| | Irresponsible $1-q$ | $-C_{x1}$ $-C_y$ $R_z$ 0 | $-C_{x1}$ $-C_y$ 0 0 | $-C_{x1}$ 0 $R_z$ 0 0 | $-C_{x1}$ 0 0 0 |
| Nonstandard operation $1-x$ | Dutiful $q$ | $-C_{x2}-F1-(1-\alpha)F2$ $-C_y+(1-\alpha)F2$ 0 $(1-\alpha)(-N_q)-C_q$ | $-C_{x2}-F1-(1-\alpha)F2$ $-C_y+(1-\alpha)F2$ 0 $-C_q$ | $-C_{x2}-\alpha F1-(1-\alpha)(Rx+\beta F3)$ $(1-\alpha)(-N_y+\beta T_y)$ $(1-\alpha)(\beta F3-N_z)$ $(1-\alpha)(-N_q)-C_q$ | $-C_{x2}-\alpha F1$ $(1-\alpha)(-N_y)$ $(1-\alpha)(-N_z)$ $-C_q$ |
| | Irresponsible $1-q$ | $-C_{x2}-F1-F2$ $F4+F2-C_y$ 0 $-N_q+F4$ | $-C_{x2}-F1-F2$ $F4+F2-C_y$ 0 $-F4$ | $-C_{x2}-R_x-\beta F3$ $-N_y+\beta T_y$ $\beta F3-N_z$ $-N_q$ | $-C_{x2}-Ny$ $-N_z$ 0 0 |

Note: from top to bottom, it is the income of the enterprise, the grass-roots government, farmers, and the village committee.

player's choices: (1) Standardized vs. Nonstandard Operation. Enterprises choose between standardized and nonstandard operations, which impacts costs, reputation, and potential fines. (2) Role of Village Committees. Village committees can either fulfill their duties or fail, influencing farmer dissatisfaction and potential fines from the government. (3) Government Supervision. The grass-roots government incurs costs for supervision and collects fines from non-compliant enterprises or village committees. (4)Farmer's Environmental Awareness. The level of environmental awareness among farmers affects their response to enterprise operations and the resulting payoffs. (5) Probabilities. Probabilities of successful coordination and continued rights protection influence the expected outcomes and costs for each player.

**4.2.1 Stability analysis of enterprise strategy.** The expected return, dynamic equation, and first-order derivative of the enterprise's choice of standard operation and nonstandard operation are respectively:

$$U_x = -C_{x1} \tag{1}$$

$$U_{1-x} = -C_{x2} - yqF1 - (1-y)q\alpha F1 - yq(1-\alpha)F2$$

$$-y(1-q)(F1+F2) - z(1-y)(1-\alpha q)(\beta F3 + R_x) \tag{2}$$

$$F(x) = d_x/d_t = x(1-x)[-C_{x1} + C_{x2} + yqF1 + (1-y)q\alpha F1 + yq(1-\alpha)F2$$

$$yq(1-\alpha)F2 + y(1-q)(F1+F2) + z(1-y)(1-\alpha q)(\beta F3 + R_x)] \tag{3}$$

$$F(x) = (1-2x)[-C_{x1} + C_{x2} + yF1 + (1-y)q\alpha F1 + y(1-\alpha q)F2$$

$$+z(1-y)(1-\alpha q)(\beta F3 + R_x)] \tag{4}$$

Most of the parameter meanings in the formula can be seen in Table 1. Besides, $U_x$ represents the expected return of the enterprise when choosing standardized operation. $U_{1-x}$ represents The expected return of the enterprise when choosing nonstandard operation. This model is part of a game-theoretic or economic framework where different stakeholders interact under various probabilistic factors and economic pressures: Enterprises must choose between standardized and nonstandard operations, considering costs, fines, investments, and reputation impacts. Village Committees may succeed or fail in coordinating environmental efforts, impacting fines and compensations. Government Supervision affects the cost dynamics through fines and incentivizes enterprises to choose compliance. Farmers' Environmental Awareness influences the overall societal response, affecting reputation and compensation costs for non-compliance.

Eqs (1) and (2) calculate the economic outcomes for enterprises based on their operational choices, factoring in the costs and fines associated with compliance or non-compliance. Eq (3) calculates the evolution of enterprise decisions over time, capturing the probability of choosing standardized operations in response to various factors. Eq (4) provides a streamlined view of decision-making dynamics, highlighting key factors that shift the balance towards or against standardized operations.

According to the stability theorem of differential equations, the enterprise's business strategy in a stable state must meet $F(x) = 0$ and $F'(x) < 0$。

**Proposition 1:** When $z > z_0$, the stability strategy of the enterprise is to standardize the operation; When $z < z_0$, the stability strategy of the enterprise is nonstandard operation; When $z =$

$z_0$, the stabilization strategy cannot be determined, and the threshold is
$z_0 = [Cx1 - C_{x2} - yF1 - (1-y)q\alpha F1 - y(1-\alpha q)F2]/(1-y)(1-\alpha q)(\beta F3 + R_x)$.

**Proof:** let

$H(z) = -C_{x1} + C_{x2} + yF1 + (1-y)q\alpha F1 + y(1-\alpha q)F2 + z(1-y)(1-\alpha q)(\beta F3 + R_x)$.
Because $H'(z)>0$, $H(z)$ is an increasing function concerning z. When $z > z_0$, $H(z) > 0$, $F(x)|_{x=1} = 0$ and $F'(x)|_{x=1}<0$, and x = 1 has stability; When $z < z_0$, $H(z) < 0$, $F(x)|_{x=0} = 0$ and $F'(x)|_{x=0}<0$, and x = 0 have stability. When $z = z_0$, $H(z) = 0$, $F(x) = 0$, and $F'(x) = 0$, $x \in [0,1]$ is stable, and it cannot determine a stabilization strategy.

Proposition 1 shows that when farmers' awareness of environmental protection is improved, and they always pay attention to the environmental quality, rather than being indifferent and letting it go, enterprises will suffer reputation losses and possible economic fines if they adhere to nonstandard management. Therefore, with the improvement of farmers' environmental protection awareness, enterprises will tend to operate in a standardized manner.

**4.2.2 Stability analysis of grass-roots government strategy.** The expected return, dynamic equation, and first-order derivative of grass-roots government's choice of regulation and non-regulation are respectively:

$$U_y = -C_y + (1-x)[F2 + (1-q)F4 - q\alpha F2] \tag{5}$$

$$U_{1-y} = (1-x)(1-\alpha q)(-N_y + z\beta T_y) \tag{6}$$

$$F(y) = \frac{dy}{dt} = y(1-y)\left(U_y - U_{1-y}\right)$$

$$= y(1-y)[-C_y + (1-x)(1-\alpha q)(F2 + N_y - z\beta T_y) + (1-x)(1-q)F4] \tag{7}$$

$$F(y) = (1-2y)[-C_y + (1-x)(1-\alpha q)(F2 + N_y - z\beta T_y) + (1-x)(1-q)F4] \tag{8}$$

According to the stability theorem of differential equations, the regulation strategy of the grass-roots government in a stable state must meet $F(y) = 0$ and $F'(y)<0$.

**Proposition 2:** when $z>z_0$, the stability strategy of the grass-roots government is not to supervise; When $z<z_0$, the stability strategy of the grass-roots government is to supervise; When $z = z_0$, the stabilization strategy cannot be determined, and the threshold is
$z_0 = (F2 + N_y)/\beta Ty - [C_y - (1-x)(1-q)F4]/(1-x)(1-\alpha q)\beta T_y$.

**Proof:** let $H(z) = -C_y + (1-x)(1-\alpha q)(F2 + N_y - z\beta T_y) + (1-x)(1-q)F4$. Because $H'(z)<0$, $H(z)$ is a subtractive function concerning z. When $z > z_0$, $H(z) < 0$, $F(y)|_{y=0} = 0$ and $F'(y)|_{y=0}<0$ y = 0 has stability; When $z < z_0$, $H(z) > 0$, $F(y)|_{y=1} = 0$ and $F'(y)|_{y=1}<0$ y = 1 has stability. When $z = z_0$, $H(z) = 0$, $F(y) = 0$, and $F'(y) = 0$ $y \in [0,1]$ is stable, and it cannot determine a stabilization strategy.

Proposition 2 shows that if the probability of farmers' awareness of environmental protection increases, the stability strategy of the grass-roots government will change from regulation to non-regulation. In other words, if farmers pay more attention to environmental protection and supervise the business behavior of enterprises and actively report to the village committee or safeguard their rights, they can affect the business decisions of enterprises, which has a positive significance for protecting the rural ecological environment. The grass-roots government also can save supervision costs, and the stabilization strategy is not supervision. On the contrary, if farmers turn a blind eye to the environmental quality of rural living space, the

government must implement supervision to avoid further pollution expansion and more significant losses.

**4.2.3 Stability analysis of farmer strategy.** The expected return, dynamic equation, and first-order derivative of farmers with and without environmental awareness are respectively:

$$U_z = xR_z + (1-x)(1-y)(1-\alpha q)(\beta F3 - N_z) \tag{9}$$

$$U_{1-z} = (1-x)(1-y)(1-\alpha q)(-N_z) \tag{10}$$

$$F(z) = \frac{dz}{dt} = z(1-z)(U_z - U_{1-z}) \tag{}$$

$$= z(1-z)[xR_z + (1-x)(1-y)(1-\alpha q)\beta F3] \tag{11}$$

$$F(z) = (1-2z)[xR_z + (1-x)(1-y)(1-\alpha q)\beta F3] \tag{12}$$

According to the stability theorem of differential equations, the strategy of farmers' awareness of environmental protection in a stable state must meet $F(z) = 0$ and $F'(z) < 0$.

**Proposition 3:** when $y > y_0$, farmers' stability strategy is to have no awareness of environmental protection; When $y < y_0$, farmers' stability strategy is to have an awareness of environmental protection; When $y = y_0$, the stabilization strategy cannot be determined, and the threshold is $y_0 = 1 + xR_z/(1-x)(1-\alpha q)\beta F3$.

**Proof:** let $H(y) = xR_z + (1-x)(1-y)(1-\alpha q)\beta F3$. Because $H'(y) < 0$, $H(y)$ is a subtractive function concerning y. When $y > y_0$, $H(y) < 0$, $F(z)|_{z=0} = 0$ and $F'(z)|_{z=0} < 0$, z = 0 has stability; When $y < y_0$, $H(y) > 0$, $F(z)|_{z=1} = 0$ and $F'(z)|_{z=1} < 0$, z = 1 has stability. When $y = y_0$, $H(y) = 0$, $F(z) = 0$, and $F'(z) = 0$, $z \in [0,1]$ is stable, and it cannot determine a stabilization strategy.

Proposition 3 shows that the increases in the probability of grass-roots government supervision will change the farmers' stabilization strategy into no environmental awareness, and the decrease in the probability of grass-roots government supervision will change the farmers' stabilization strategy into environmental protection awareness. Therefore, when the government's supervision is strong enough, the government can directly influence or even change the business behavior of enterprises without requiring farmers to enhance their environmental awareness and join the supervision team.

**4.2.4 Stability analysis of village committee strategy.** The expected return, dynamic equation, and first-order derivative of the village committee fulfill duties or not:

$$U_q = x(-C_q) + (1-x)[-C_q + z(1-\alpha)(-N_q)] \tag{13}$$

$$U_{1-q} = (1-x)[z(-N_q) + y(-F4)] \tag{14}$$

$$F(q) = \frac{dq}{dt} = q(1-q)\left(U_q - U_{1-q}\right) = q(1-q)\left[-C_q + (1-x)(z\alpha N_q + yF4)\right] \tag{15}$$

$$F(q) = (1-2q)[-C_q + (1-x)(z\alpha N_q + yF4)] \tag{16}$$

According to the stability theorem of differential equations, the strategy of farmers' awareness of environmental protection in a stable state must meet $F(q) = 0$, and $F'(q) < 0$.

**Proposition 4:** when $x > x_0$, the village committee's stabilization strategy is not fulfilling duties; When $x = x_0$, the stabilization strategy cannot be determined, and the threshold is $x_0 = 1 - C_q/(z\alpha N_q + yF4)$.

**Proof:** let $H(x) = -C_q + (1-x)(z\alpha N_q + yF4)$. Because $H'(x) < 0$, $H(x)$ is a subtractive function concerning x. When $x > x_0$, $H(x) < 0$, $F(q)|_{q=0} = 0$ and $F'(q)|_{q=0} < 0$, $q = 0$ has stability; When $x < x_0$, $H(x) > 0$, $F(q)|_{q=1} = 0$ and $F'(q)|_{q=1} < 0$, $q = 1$ has stability. When $x = x_0$, $H(x) = 0$, $F(q) = 0$, and $F'(q) = 0$, $q \in [0,1]$ is stable, it cannot determine a stabilization strategy.

Proposition 4 shows that the increased probability of the standardized enterprise operation will change the enterprise's stabilization strategy into not fulfilling duties. The decreased probability of the standardized enterprise operation will change the enterprise's stabilization strategy into fulfilling duties. When the enterprise operates standardized, there will be no contradiction between the enterprise and the farmers in terms of pollution. The village committee naturally does not need to act as a bridge between the enterprise and the farmers to coordinate the contradictions between the two sides.

## 4.3 Stability analysis of combined strategy

The stability of 16 strategic equilibrium points in the quadripartite evolutionary game is analyzed. According to the dynamic replication equation of each game subject, the Jacobian matrix of the dynamic replication system is obtained as follows:

$$J = \begin{bmatrix} \partial F(x)/\partial x & \partial F(x)/\partial y & \partial F(x)/\partial z & \partial F(x)/\partial q \\ \partial F(y)/\partial x & \partial F(y)/\partial y & \partial F(y)/\partial z & \partial F(y)/\partial q \\ \partial F(z)/\partial x & \partial F(z)/\partial y & \partial F(z)/\partial z & \partial F(z)/\partial q \\ \partial F(q)/\partial x & \partial F(q)/\partial y & \partial F(q)/\partial z & \partial F(q)/\partial q \end{bmatrix} \tag{17}$$

Bring $(0,0,0,0)$ into the Jacobian matrix:

$$J = \begin{bmatrix} -C_{x1} + C_{x2} & 0 & 0 & 0 \\ 0 & -C_y + F2 + F4 + N_y & 0 & 0 \\ 0 & 0 & \beta F3 & 0 \\ 0 & 0 & 0 & -C_q \end{bmatrix} \tag{18}$$

The specific eigenvalues $\lambda_1, \lambda_2, \lambda_3, \lambda_4$ can be obtained by substituting 16 local equilibrium points into the Jacobian matrix. The asymptotic stability analysis of the equilibrium points of the dynamic replication system is shown in Table 3:

Condition A: $-C_{x1} + C_{x2} + R_x + \beta F3 < 0$, $-C_y + F2 + F4 + N_y - \beta T_y < 0$, $-Cq + \alpha Nq < 0$

Condition B: $C_{x1} - C_{x2} + R_x + \beta F3 > 0$

Condition C: $-C_{x1} - C_{x2} + \alpha F1 + R_x + \beta F3 < 0$, $-C_y + (1-\alpha)(F2 + N_y - \beta T_y) < 0$, $-Cq + \alpha Nq > 0$

According to Table 3, there are three possible equilibrium points: $(0,0,1,0)$, $(0,0,1,1)$ and $(1,0,1,0)$, which are analyzed respectively as follows:

When condition A is established, $(0,0,1,0)$ becomes the equilibrium point, and the corresponding strategy is (nonstandard, non-regulatory, environmental awareness, irresponsible). ① By $-C_{x1} - C_{x2} + R_x + \beta F3 < 0$ deducing $C_{x1} - C_{x2} > R_x + \beta F3$, the cost saved by the nonstandard operation of the enterprise is greater than the sum of its reputation loss and economic compensation that may need to be paid to farmers. In this case, enterprises will gain more extra income due to nonstandard operations. Even if farmers have environmental awareness and

**Table 3. Asymptotic stability analysis of the equilibrium point of replication dynamic system.**

| Equilibrium point | Characteristic value | Positivity/negativity | Stability |
|---|---|---|---|
| $(0,0,0,0)$ | $-C_{x1}+C_{x2},-C_y+F2+F4+N_y\beta F3,-C_q$ | (-, U, +, -) | Instability |
| $(1,0,0,0)$ | $-(-C_{x1}+C_{x2}),-C_y,R_z,-C_q$ | (+, -, +, -) | Instability |
| $(0,1,0,0)$ | $-C_{x1}+C_{x2}+F2,-(-C_y+F2+F4+N_y),0,-C_q+F4$ | (U, U, 0, +) | Instability |
| **$(0,0,1,0)$** | $\mathbf{-C_{x1}+C_{x2}+R_x+\beta F3,-C_y+F2+F4+N_y-\beta T_y,-\beta F3,-C_q+\alpha Nq}$ | **(U, U, -, U)** | **ESS, when condition a is met** |
| $(0,0,0,1)$ | $-C_{x1}+C_{x2}+\alpha F1,-C_y+(1-\alpha)(F2+N_y),(1-\alpha)\beta F3,C_q$ | (U, -, +, +) | Instability |
| $(1,1,0,0)$ | $-(-C_{x1}+C_{x2}+F1+F2),C_y,R_z,-C_q$ | (U, +, +, -) | Instability |
| **$(1,0,1,0)$** | $\mathbf{-(-C_{x1}+C_{x2}+R_x+\beta F3),-C_y,-R_z,-C_q}$ | **(U, -, -, -)** | **ESS, when condition b is met** |
| $(1,0,0,1)$ | $-(-C_{x1}+C_{x2}+\alpha F1),-C_y,R_z,C_q$ | (U, -, +, +) | Instability |
| $(0,1,0,1)$ | $-C_{x1}+C_{x2}+F1+(1-\alpha)F2,-[-C_y+(1-\alpha)(F2+N_y)],0,-(-C_q+F4)$ | (U, +, 0, +) | Instability |
| $(0,1,1,0)$ | $-C_{x1}+C_{x2}+F1+F2,-(-C_y+F2+F4+N_y-\beta T_y),0,-C_q+\alpha N_q+F4$ | (U, U, 0, +) | Instability |
| **$(0,0,1,1)$** | $\mathbf{-C_{x1}+C_{x2}+\alpha F1+Rx+\beta F3,-C_y+(1-\alpha)(F2+N_y-\beta T_y),-(1-\alpha)\beta F3,}$ $\mathbf{-(-C_q+\alpha N_q)}$ | **(U, U, -, U)** | **ESS, when condition c is met** |
| $(1,1,1,0)$ | $-(-C_{x1}+C_{x2}F1+F2),C_y,-R_z,C_q,-C_q$ | (U, +, -, -) | Instability |
| $(1,1,0,1)$ | $-(-C_{x1}+C_{x2}F1+F2),C_y,R_z,C_q$ | (U, +, +, +) | Instability |
| $(1,0,1,1)$ | $-[-C_{x1}+C_{x2}+\alpha F1+(1-\alpha)(R_x+\beta F3)],-C_y,-R_z,C_q$ | (U, -, 0, +) | Instability |
| $(0,1,1,1)$ | $-C_{x1}+C_{x2}+F1+(1-\alpha)F2,-[-C_y+(1-\alpha)(F2+N_y-\beta T_y)],0,-(-C_q+\alpha N_q+F4)$ | (U, U, 0, +) | Instability |
| $(1,1,1,1)$ | $-(-C_{x1}+C_{x2}+F1+(1-\alpha)F2),C_y,-R_x,C_q$ | (U, +, -, +) | Instability |

Note: "U" means the positive and negative symbols are uncertain

protect their rights, enterprises will still choose nonstandard operations. ② By $-C_y+F2+F4+N_y-\beta T_y<0$ deducing $-C_y+F2+F4<\beta T_y-N_y$, if the grass-roots government chooses to supervise, its regulatory income is less than 0, which will not increase revenue but will bring losses. At the same time, the absolute value of this loss is greater than that of the loss without supervision. At this time, for economic reasons, the grass-roots government will choose not to supervise. ③ By $-C_q+\alpha Nq<0$ deducing $Nq<C_q+(1-\alpha)Nq$, the loss caused by the village committee failing to do its duty is less than that caused by the village committee doing its duty but coordination failure. The village committee chose not to do its duty to avoid more losses.

When condition C is established, $(0,0,1,1)$ becomes the equilibrium point, and the corresponding strategy is (nonstandard, non-regulatory, environmental awareness, dutiful). ① By $-C_{x1}+C_{x2}+\alpha F1+R_x+\beta F3<0$ deducing $C_{x1}-C_{x2}-\alpha F1>R_x+\beta F3$ and $C_{x1}-C_{x2}>R_x+\beta F3$, the cost saved nonstandard operation of the enterprise, even after deducting the additional investment in environmental protection equipment, is still more significant than the sum of the reputation loss and the economic compensation of the enterprise. Whether the enterprise accepts or does not accept the coordination of the village committee, the additional income from the nonstandard operation is enough to pay potential reputation losses and possible economic compensation. At this time, the enterprise will choose nonstandard operations. ② By $-C_y+(1-\alpha)(F2+N_y-\beta T_y)<0$ deducing $-C_y+(1-\alpha)F2<(1-\alpha)(\beta T_y-N_y)$, when the village committee fails to coordinate, if the grass-roots government chooses to supervise, its supervision income is less than 0, then the grass-roots government will choose not to supervise. ③ By $C_q+\alpha Nq>0$ deducing $Nq>C_q+(1-\alpha)Nq$, the loss caused by the village committee's failure to do its duty is greater than the cost caused by the coordination failure. In other words, when the village committee chooses to be dutiful, even if the coordination fails, the loss it suffers is smaller than that of not being dutiful. By comparison, the village committee tends to be

dutiful. When $\alpha \rightarrow 1$, the village committee has an excellent probability of successful coordination. At this time, the village committee tends to be dutiful, and the equilibrium point gradually transforms from (0,0,1,0) to the equilibrium point (0,0,1,1).

When condition B is established, (1,0,1,0) becomes the equilibrium point, and the corresponding strategy is (standard, non-regulatory, environmental awareness, irresponsible). By $-C_{x1}-C_{x2}+R_x+\beta F3>0$ deducing $C_{x1}-C_{x2}<R_x+\beta F3$, the cost saved by the nonstandard operation of the enterprise is less than the sum of the reputation loss of the enterprise and the economic compensation that may need to be paid to farmers. Suppose the farmers with environmental awareness safeguard their rights, and the cost saved by the enterprise's nonstandard operation, or the additional income it brings, cannot cover the reputation loss and possible economic compensation. In that case, the enterprise will operate in a standardized manner at this time.

The reputation mechanism and rights protection mechanism under the premise of farmers' awareness of environmental protection can make the stability strategy of enterprises standardize operation. When condition B is met, improving reputation loss can also improve the probability of safeguarding rights and getting economic compensation. Let $Rx+\beta F3>C_{x1}-C_{x1}$. In this case, the game system has only one ideal stable strategy combination (1,0,1,0), and if condition A and condition C are not satisfied, then (0,0,1,0) and (0,0,1,1) are unstable. Therefore, increasing the reputation loss caused by enterprises' nonstandard operations is significant to maintaining rural environmental quality. Improving enterprises' economic compensation to farmers is necessary to ensure rural environmental quality, and farmers' awareness of environmental protection and rights protection should also be strengthened.

## 5 Model simulation

This paper uses MATLAB software to conduct numerical simulations on the evolutionary trajectory of each game party, which can more intuitively show the key elements that influence the evolution process and results of multi-party games. The parameters are set as follows:

1. The enterprise enters the countryside, facing environmental pollution in the production process, it will purchase pollution treatment facilities or choose green production approaches when operating in a standardized manner, and the cost is $C_{x1} = 10$, and bring positive utility $R_z = 10$ to farmers with environmental awareness.

2. When the enterprise chooses nonstandard operation, the operating cost is $C_{x2} = 2$.

3. If the village committee dutifully exhorts the enterprise's nonstandard business behavior, it will pay $C_q = 4$, and the enterprise obeys the advice and adds $F1 = 8$ to the investment in environmental protection facilities.

4. The regulatory cost of the grass-roots government is $C_y = 12$. For enterprises that do not operate in a standardized manner, the grass-roots government will impose a fine of $F2 = 15$ once it is found. A penalty $F4 = 6$ will be charged to the village committee that irresponsibly handles the environmental damage of enterprises simultaneously.

5. The continuous nonstandard operations of enterprises will bring losses to farmers and society if $N_z = 8$ and $N_y = 15$. The negative effect of farmers with environmental awareness on the village committee who do not do their duty is $N_q = 5$, and the reputation loss of enterprises is $R_x = 6$. The probability that farmers choose to protect rights is $\beta = 0.1$, the economic compensation obtained is $F3 = 7$, and the recovered social loss is $T_y = 10$.

6. The initial strategies of each player are $x = 0.4, y = 0.3, z = 0.2, q = 0.3$.

More details can be found in S1 File.

## 5.1 Influence of the coordination ability of the village committee

Let $\alpha = \{0.2, 0.5, 1\}$, the evolution process and results of the strategies of the four-party game players are shown in Figs 2–4:

It can be seen that the improvement of the coordination ability of the village committee can reduce the probability of government supervision but cannot promote the standardized operation of enterprises. When $\alpha$ is at a low level, the coordination of the village committee is likely to fail. Not only are farmers dissatisfied, but the coordination cost of the premise has also become a silent cost. Therefore, the village committee will tend to choose not to be dutiful, and the probability of enterprises' nonstandard operation will also rise. The grass-roots government will also increase supervision. With the continuous improvement of $\alpha$, the likelihood of successful coordination of the village committee also increases. Hence, it gradually becomes

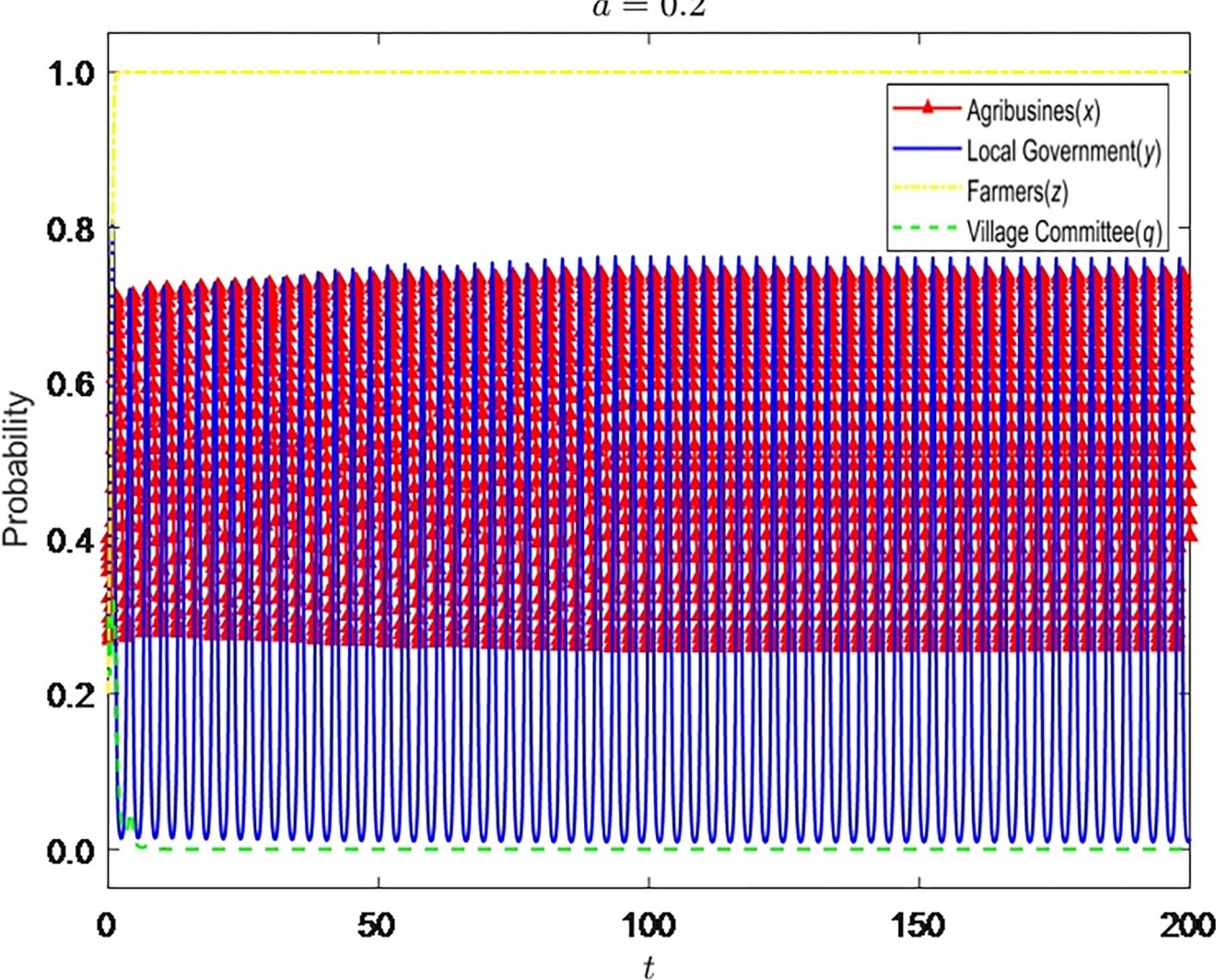

**Fig 2. Evolution process and results of the strategies of the four-party game players ($\alpha = 0.2$).**

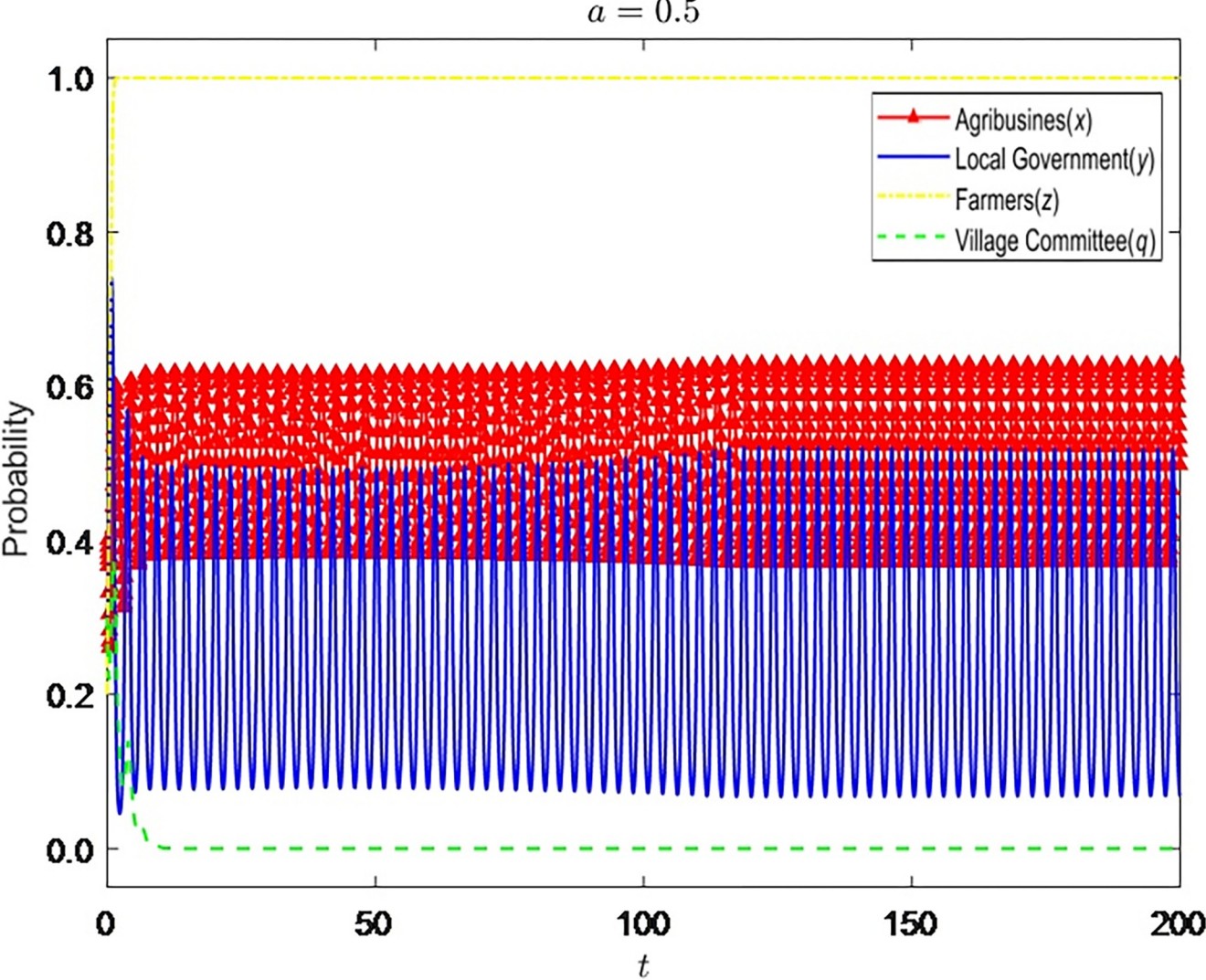

**Fig 3. Evolution process and results of the strategies of the four-party game players ($\alpha = 0.5$).**

dutiful and stops enterprises' nonstandard business behavior in time. At this time, the grass-roots government will also reduce the probability of supervision and slowly change to the strategy of non-supervision. However, due to the limited rights of the village committee, even if it has a strong coordination ability, it can only persuade enterprises to supplement the input of environmental protection equipment, and it is difficult to implement other punishment measures, so the village committee cannot have a significant impact on the business decisions of enterprises.

## 5.2 Influence of farmers' environmental protection awareness and safeguarding rights probability

Let z = {0,0.7} and the strategy evolution process and results of the other three parties are shown in Figs 5–7:

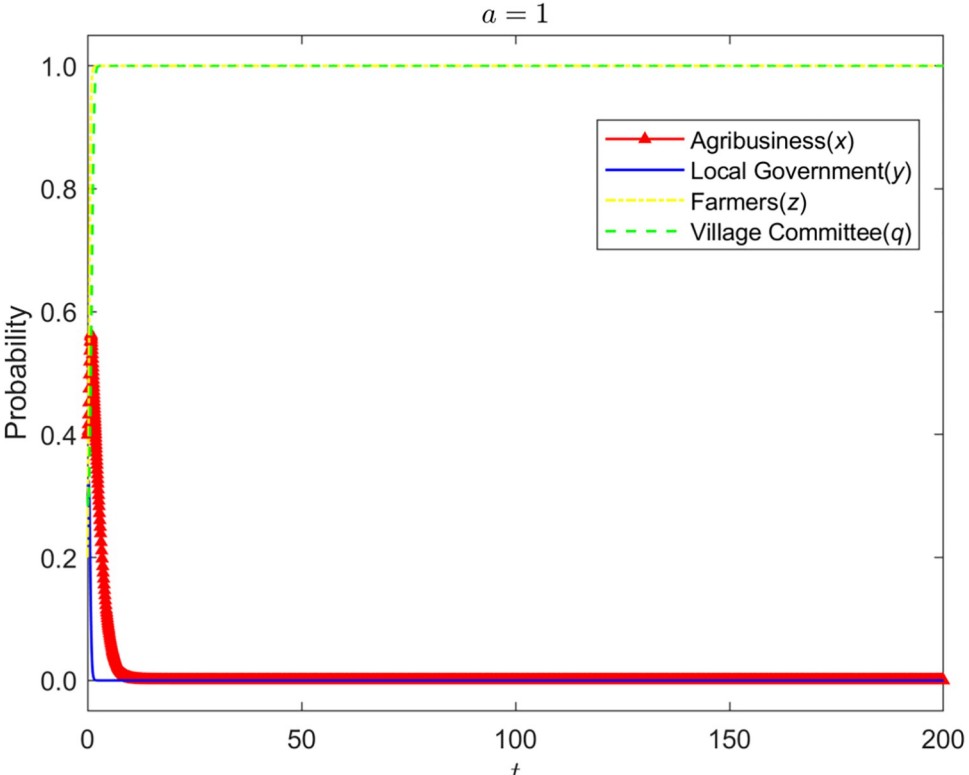

**Fig 4. Evolution process and results of the strategies of the four-party game players ($\alpha = 1$).**

It can be seen that farmers' lack of awareness of environmental protection has a server negative impact on rural environmental protection. When farmers have no environmental protection awareness, enterprises and village committees have no supervision from farmers, and enterprises will gradually stabilize in nonstandard operation. The supervision strategy of the grass-roots government is in an unstable state, and the village committee will also change its strategy with the change of the government's supervision strategy. When farmers are aware of environmental protection, it has the most significant impact on the strategic choice of the village committee. To avoid farmers' dissatisfaction, the village committee will continue to be dutiful. The dual participation of farmers and village committees has made the grass-roots government relax supervision.

Nevertheless, the village committee has less authority to manage the enterprise, and enterprises are still stable in nonstandard operation. On the premise of farmers' awareness of environmental protection, the improvement of rights protection probability can make up for the shortcomings of poor management of village committees. The increase in the probability of farmers' rights protection has promoted the transformation of the stability strategy of enterprises from nonstandard operation to standard operation. After the enterprises choose standard operation as the stability strategy, the stability strategy of village committees and grass-roots governments has changed to be irresponsible and non-supervision.

### 5.3 Influence of enterprise's reputation loss

Let $Rx = \{6, 8, 10\}$, the evolution process and results of the strategy of the four-party game are shown in Figs 8–10:

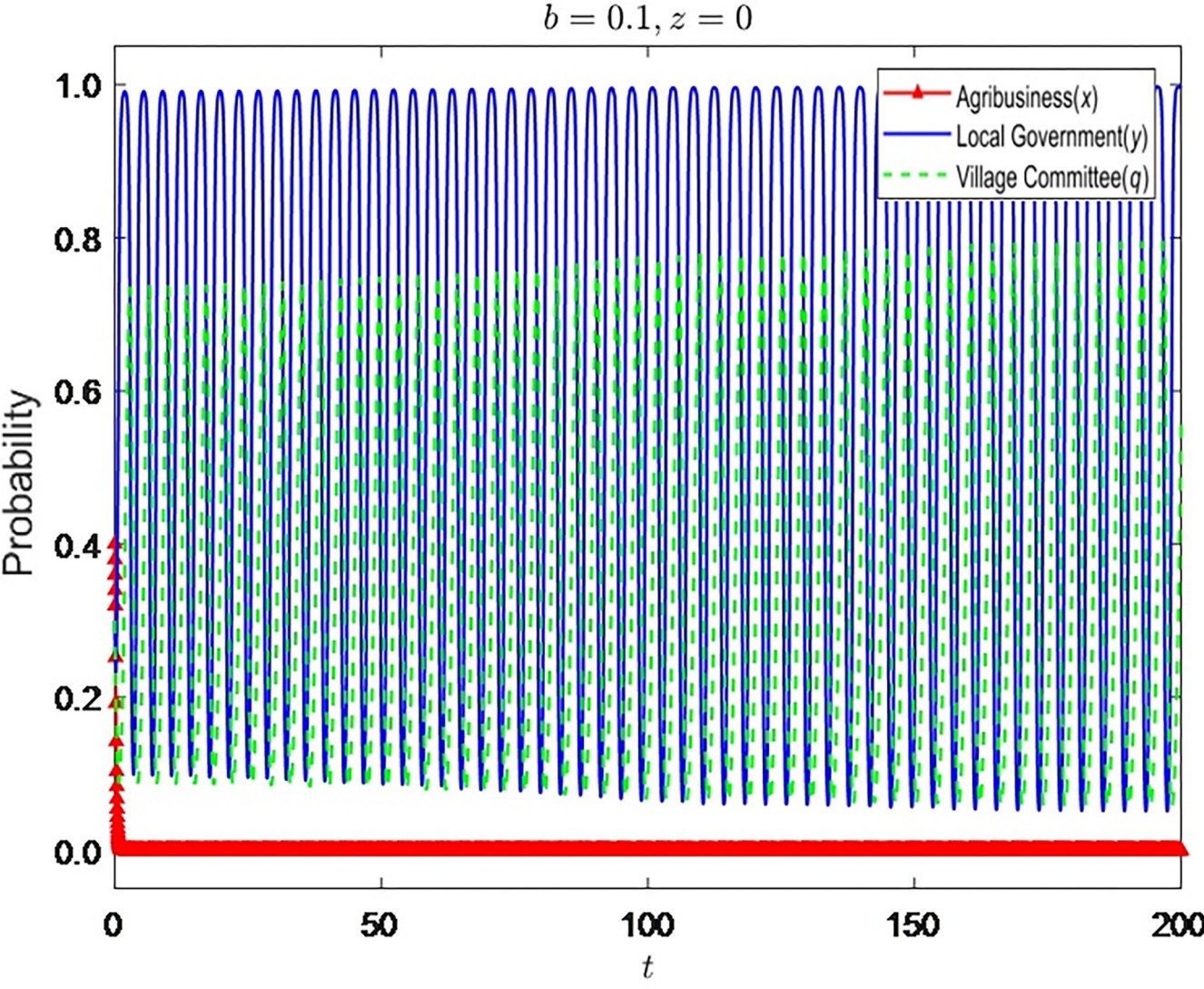

**Fig 5. Strategy evolution process and results of the other three parties ($b = 0.1$, $z = 0$).**

It can be seen that the loss of corporate reputation is affected by farmers' awareness of environmental protection, and the increase in reputation loss promotes the standardized operation of enterprises. Environmental awareness is the basis of reputation value. Only farmers with environmental awareness will deeply hate enterprises' behavior of polluting the environment and have a wrong impression on enterprises, resulting in the enterprises' reputation loss. When the reputation loss is slight, enterprises tend to operate nonstandard. Still, farmers' environmental protection awareness led the village committee to adopt the strategy of fulfilling duties, and the grass-roots government adopted the approach of non-supervision. With the continuous increase of reputation loss, the design of enterprises has changed to standardized operation. In this process, the village committee supervises and exhorts the enterprise. The village committee will not withdraw from the management of the enterprise until the enterprise stabilizes in the standardized business strategy. When the village committee plays a role, non-supervision is always the stability strategy of the grass-roots government.

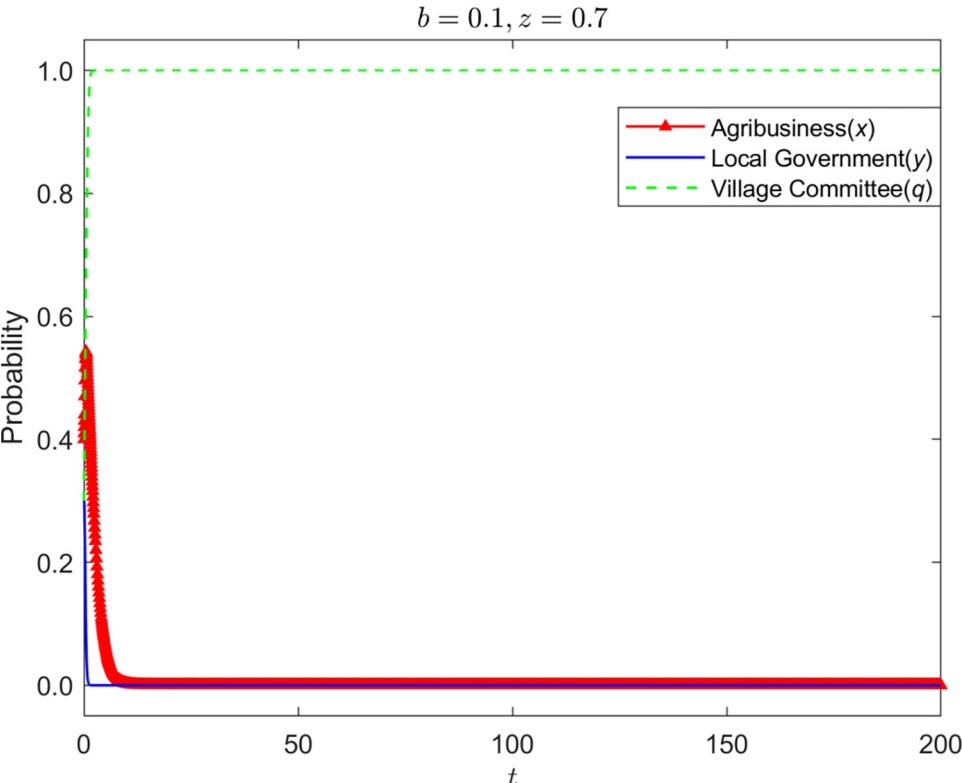

**Fig 6. Strategy evolution process and results of the other three parties ($b = 0.1, z = 0.7$).**

## 6 Conclusion

In the context of the Rural Revitalization Strategy, more and more enterprises come to rural areas to develop industries. Still, the nonstandard operations of some enterprises also pollute the rural environment. Therefore, this paper establishes a dynamic game model among the village committee, grass-roots government, enterprises, and farmers and studies the impact of the village committee, grass-roots government, and farmers on an enterprise's operation strategy. The main conclusions are shown as follows:

1. Village committee is challenging to promote positive environmental behavior of enterprises, which requires grass-roots government supervision. Since the village committee can only exhort the enterprise and has no right to impose mandatory punishment, the enterprise can choose to accept or not. Moreover, even if the enterprise agrees with the exhortation of the village committee, it can still obtain income from nonstandard operations while adding environmental protection equipment. Therefore, under normal circumstances, enterprises tend to operate irregularly. The limited administrative power of the village committee determines its weak binding force on enterprises, which leads to the grass-roots government using supervision means to force enterprises to change into standardized operation strategies. Compared with the village committee, the supervision means of the grass-roots government are more diverse and comprehensive. They can not only implement the whole process supervision, but also choose mild means such as interviews and exemption from punishment or authoritarian means such as legislation, economic punishment, and administrative coercion. The grass-roots government is more restrictive and

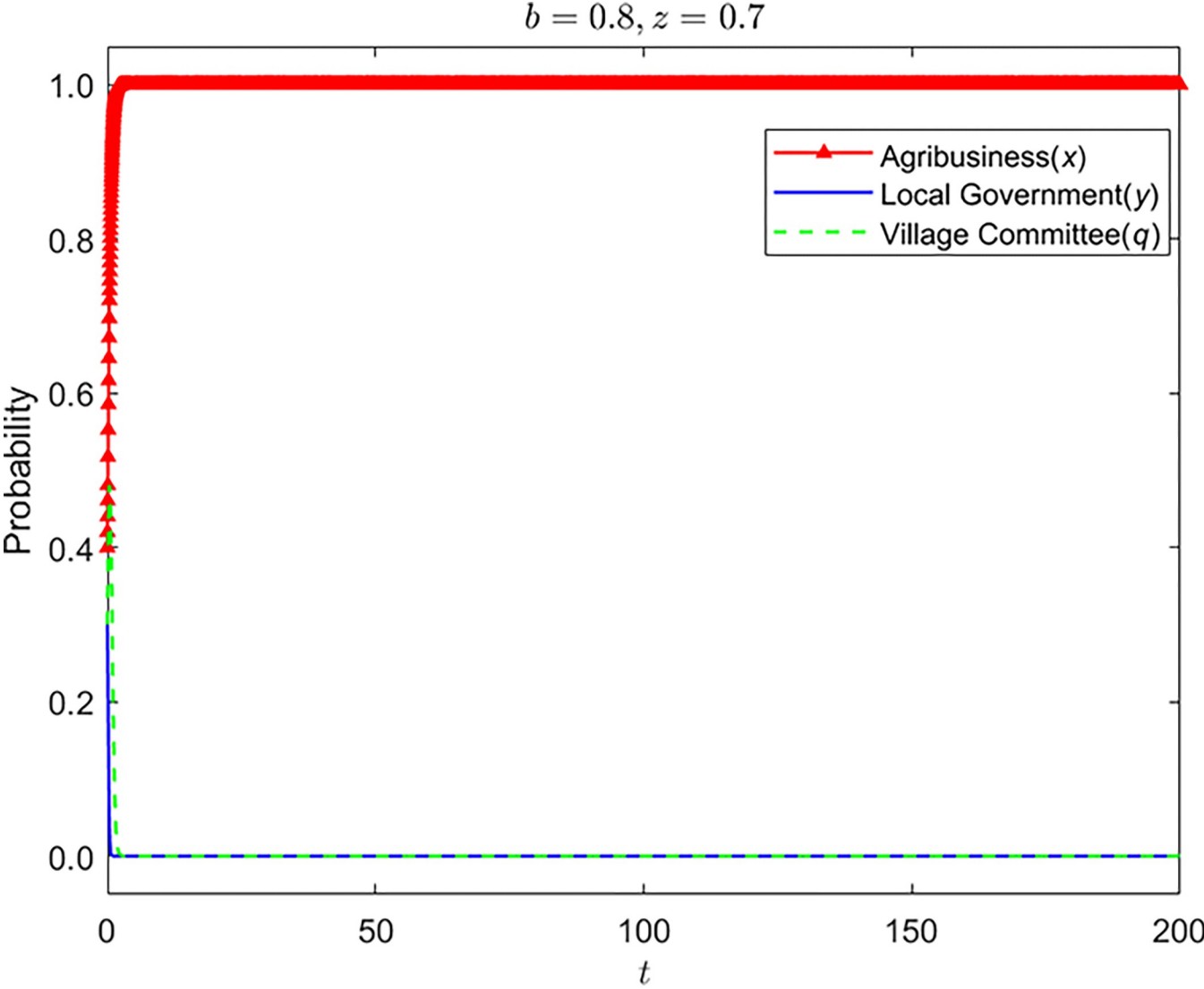

**Fig 7. Strategy evolution process and results of the other three parties ($b = 0.8, z = 0.7$).**

deterrent to enterprises and can effectively enlarge the preventive effect on enterprises' environmental pollution behavior.

2. Improving the coordination ability of village committees is conducive to reducing the supervision burden of grass-roots government. Although the village committee cannot directly interfere with the business strategy of the enterprise, if the village committee has a high coordination ability, it can prevent the further deterioration of the enterprise's environmental pollution behavior, thereby reducing the cost of the grass-roots government to deal with the malignant events of environmental pollution. Therefore, improving the coordination ability of village committees has a positive significance for reducing the social losses of environmental pollution and reducing the supervision burden of grass-roots governments.

The ability of the village cadres limits the coordination ability of the village committee. When village cadres have strong personal abilities, they can communicate with enterprises

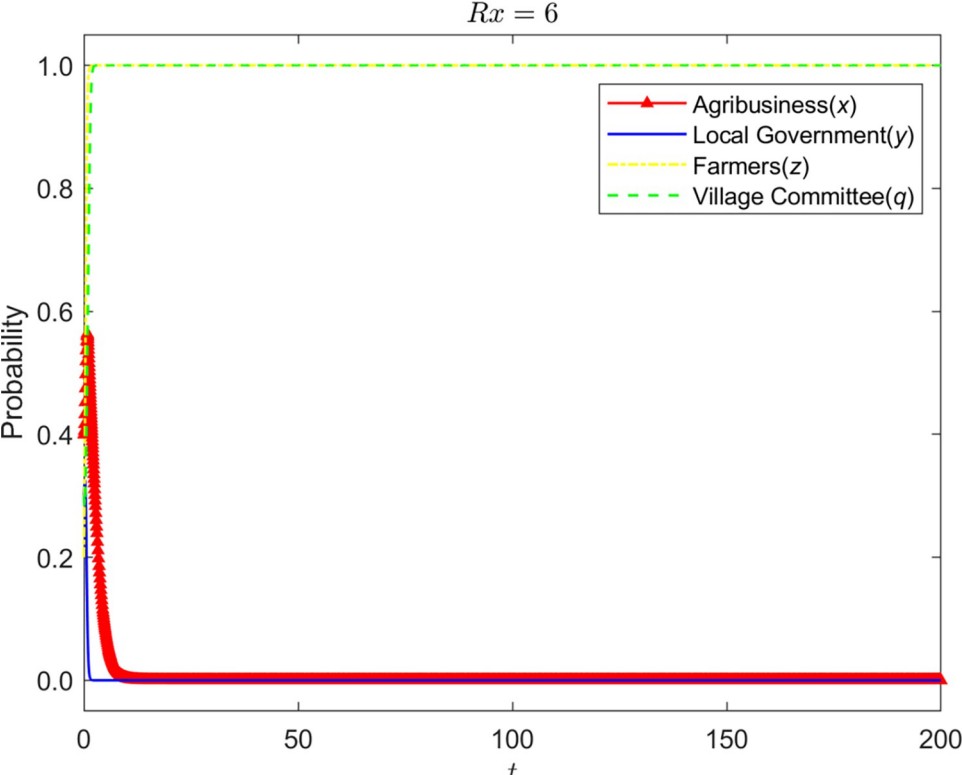

**Fig 8. Evolution process and results of the strategy of the four-party game ($Rx = 6$).**

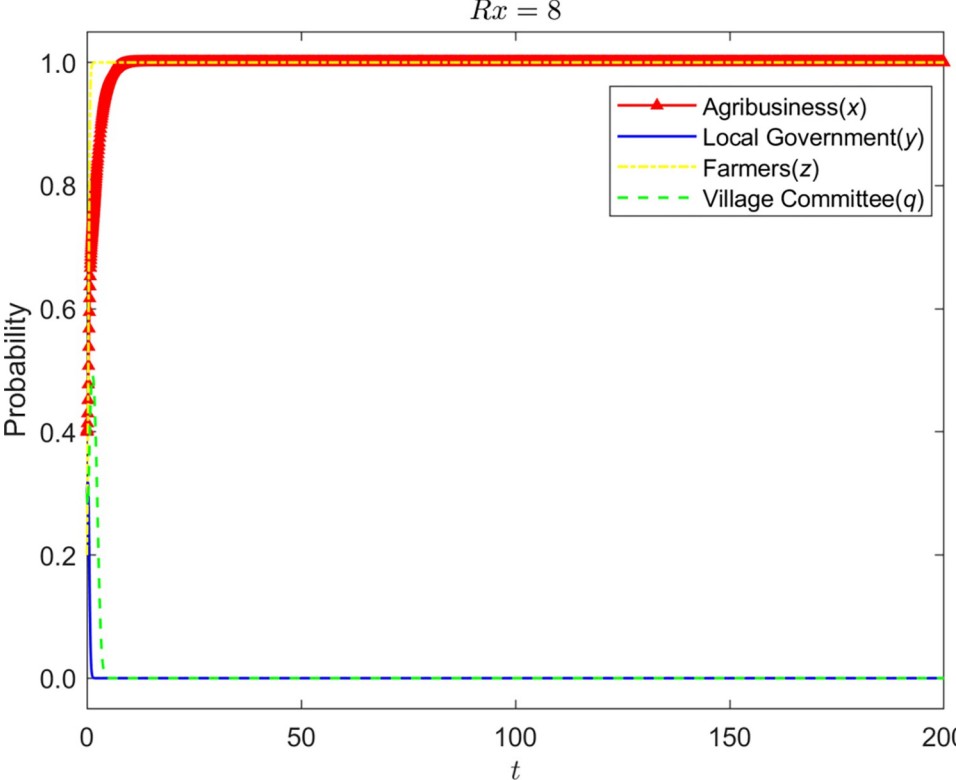

**Fig 9. Evolution process and results of the strategy of the four-party game ($Rx = 8$).**

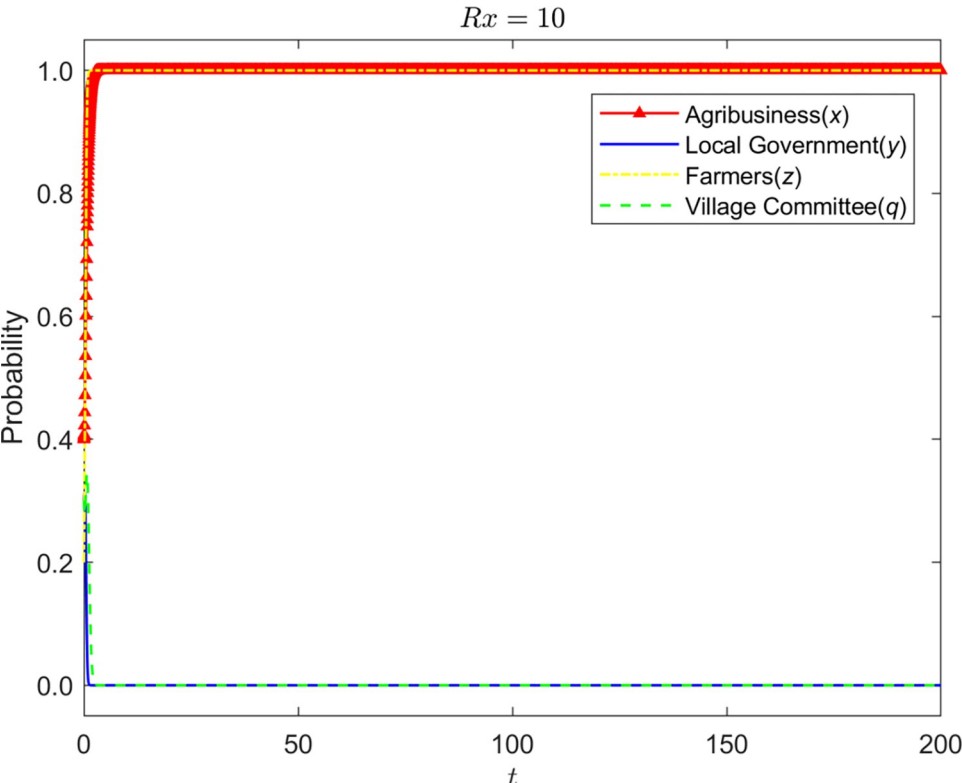

**Fig 10. Evolution process and results of the strategy of the four-party game (*Rx* = 10).**

through their excellent communication, negotiation, and transaction processing skills to maximize the environmental benefits for rural areas and farmers. Therefore, on the one hand, improving the treatment of village cadres can attract outstanding management talents and high-quality talents in the society to enter the village committee; On the other hand, the village committee should also strengthen the training of current village cadres, and carry out skills training such as communication skills, negotiation skills, and speech skills, to improve their abilities.

3. Farmers with environmental awareness affect the environmental behavior of enterprises through the rights protection mechanism and reputation mechanism. Farmers' awareness of environmental protection is necessary for the effectiveness of the rights protection mechanism and reputation mechanism. China's main rural environmental governance body is the grass-roots government. Although farmers are the most direct stakeholder, they are often absent in environmental governance. The environment is a particular public good, and its public nature makes farmers less motivated to maintain and more willing to refuse others to "free ride", resulting in the minimum participation of farmers. However, farmers' awareness of environmental protection is the premise of actively safeguarding their rights and giving play to the reputation mechanism. Only in this way can farmers pay attention to the environmental quality and whether enterprises have environmental pollution behaviors. Once farmers find the damage behaviors of enterprises to the rural environment, they can immediately take action to protect their rights and try to stop it in time when the environmental pollution has just occurred, or the loss is relatively small, instead of waiting for

further deterioration of environmental pollution. Farmers and villages need to spend more money to compensate for the damage at that time.

Therefore, cultivating farmers' environmental protection awareness can effectively affect enterprises' environmental behavior. Nevertheless, only having environmental awareness cannot form enough deterrents to the business decision-making of enterprises. A rights protection mechanism and a reputation mechanism must be combined. On the premise of improving farmers' awareness of environmental protection, unblocking rights protection channels and reputation feedback channels can effectively affect the choice of business strategies of enterprises.

## Supporting information

**S1 File.**
(DOCX)

## Author Contributions

**Conceptualization:** Xujun Zhai, Hong Lin.

**Funding acquisition:** Xujun Zhai.

**Investigation:** Lian Zheng.

**Methodology:** Xujun Zhai.

**Resources:** Lian Zheng.

**Software:** Lian Zheng.

**Writing – original draft:** Xujun Zhai.

**Writing – review & editing:** Hong Lin.

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
