## [Decision Letter · Decision Letter 0]

5 Jul 2024

PONE-D-23-32398Four-party evolutionary game analysis of enterprise environmental behaviorPLOS ONE

Dear Dr. Lin,

Thank you for submitting your manuscript to PLOS ONE. After careful consideration, we feel that it has merit but does not fully meet PLOS ONE’s publication criteria as it currently stands. Therefore, we invite you to submit a revised version of the manuscript that addresses the points raised during the review process.

I strongly suggest you to follow the detailed reviewer's comments and suggestions in order to imporve the quality of your manuscript and proceed with the publication process.

We look forward to receiving your revised manuscript.

Kind regards,

Marcelo Dionisio

Academic Editor

PLOS ONE

Journal Requirements:

"Xujun Zhai, National Social Science Foundation of China (19BJL054)."

Reviewers' comments:

Reviewer's Responses to Questions

**Comments to the Author**

1. Is the manuscript technically sound, and do the data support the conclusions?

Reviewer #1: Yes

2. Has the statistical analysis been performed appropriately and rigorously? 

Reviewer #1: N/A

3. Have the authors made all data underlying the findings in their manuscript fully available?

Reviewer #1: Yes

4. Is the manuscript presented in an intelligible fashion and written in standard English?

Reviewer #1: Yes

5. Review Comments to the Author

Reviewer #1: In general the study is important and present a contribution. However, it needs some adjustments to be excellent:

Please explain equations in Table two and could you extract it.

Please display the meaning of each symbol in the paper. For example, the symbol U in Equation one was not presented.

It’s not obvious how did you extract equations 1, 2, 3, 4, please

By the same way, you should explain the other equations in the study.

In all figures, please don’t display the explanation in Chinese language. Explain all what to say in English.

6. PLOS authors have the option to publish the peer review history of their article (what does this mean?). If published, this will include your full peer review and any attached files.

Reviewer #1: No

---

## [Author Response · Author response to Decision Letter 0]

8 Aug 2024

Response to Reviewers

On behalf of all the contributing authors, I would like to express our sincere appreciation for the editor and reviewers’ constructive comments concerning our article. These comments are all valuable and helpful for improving our writing. Point-by-point responses to the excellent editor and two lovely reviewers are listed below this letter.

Reviewer #1

Dear Professor, thank you for your severe and constructive comments. Those comments are all valuable and very helpful for revising and improving our article, as well as the importance of guiding significance to our research.

No. Comments Response

1 Please explain equations in Table two and could you extract it. The authors add explanation for Table two.

2 Please display the meaning of each symbol in the paper. For example, the symbol U in Equation one was not presented. The authors add explanation for symbol of equations.

3 It’s not obvious how did you extract equations 1, 2, 3, 4, please

By the same way, you should explain the other equations in the study. The authors add explanation for equations in the paper.

4 In all figures, please don’t display the explanation in Chinese language. Explain all what to say in English. The authors revise all figures in English language.

Once again, thank you very much for your comments and suggestions. We appreciate the Editors/Reviewers’ warm work earnestly and hope the correction will meet with approval.

---

## [Decision Letter · Decision Letter 1]

28 Aug 2024

Four-party evolutionary game analysis of enterprise environmental behavior

PONE-D-23-32398R1

Dear Dr. Lin,

We’re pleased to inform you that your manuscript has been judged scientifically suitable for publication and will be formally accepted for publication once it meets all outstanding technical requirements.

Kind regards,

Kasi Eswarappa

Academic Editor

PLOS ONE

Additional Editor Comments (optional):

Based on the revisions and recommendation by the reviewer, I am accepting the paper for publication.

Reviewers' comments:

Reviewer's Responses to Questions

**Comments to the Author**

1. If the authors have adequately addressed your comments raised in a previous round of review and you feel that this manuscript is now acceptable for publication, you may indicate that here to bypass the “Comments to the Author” section, enter your conflict of interest statement in the “Confidential to Editor” section, and submit your "Accept" recommendation.

Reviewer #1: All comments have been addressed

2. Is the manuscript technically sound, and do the data support the conclusions?

Reviewer #1: Yes

3. Has the statistical analysis been performed appropriately and rigorously? 

Reviewer #1: Yes

4. Have the authors made all data underlying the findings in their manuscript fully available?

Reviewer #1: Yes

5. Is the manuscript presented in an intelligible fashion and written in standard English?

Reviewer #1: Yes

6. Review Comments to the Author

Reviewer #1: Many thanks for responding to comments. The author considered all the suggested changes.

Best regards

7. PLOS authors have the option to publish the peer review history of their article (what does this mean?). If published, this will include your full peer review and any attached files.

Reviewer #1: No

---

## [Editor Report · Acceptance letter]

30 Sep 2024

PONE-D-23-32398R1 

PLOS ONE

Dear Dr. Lin, 

I'm pleased to inform you that your manuscript has been deemed suitable for publication in PLOS ONE. Congratulations! Your manuscript is now being handed over to our production team.

Kind regards, 

on behalf of

Dr. Kasi Eswarappa 

Academic Editor

PLOS ONE